# The Digital Centaur as a Type of Technologically Augmented Human in the AI Era: Personal and Digital Predictors

**DOI:** 10.3390/bs15111487

**Published:** 2025-10-31

**Authors:** Galina U. Soldatova, Svetlana V. Chigarkova, Svetlana N. Ilyukhina

**Affiliations:** Faculty of Psychology, Lomonosov Moscow State University, Moscow 119991, Russia; soldatova.galina@gmail.com (G.U.S.); svetla.iluhina@gmail.com (S.N.I.)

**Keywords:** digital centaur, artificial intelligence, technologically augmented personality, emotional intelligence, digital competence, extended mind, human–AI collaboration

## Abstract

Industry 4.0 is steadily advancing a reality of deepening integration between humans and technology, a phenomenon aptly described by the metaphor of the “technologically augmented human”. This study identifies the digital and personal factors that predict a preference for the “digital centaur” strategy among adolescents and young adults. This strategy is defined as a model of human–AI collaboration designed to enhance personal capabilities. A sample of 1841 participants aged 14–39 completed measures assessing digital centaur preference and identification, emotional intelligence (EI), mindfulness, digital competence, technology attitudes, and AI usage, as well as AI-induced emotions and fears. The results indicate that 27.3% of respondents currently identify as digital centaurs, with an additional 41.3% aspiring to adopt this identity within the next decade. This aspiration was most prevalent among 18- to 23-year-olds. Hierarchical regression showed that interpersonal and intrapersonal EI and mindfulness are personal predictors of the digital centaur preference, while digital competence, technophilia, technopessimism (inversely), and daily internet use emerged as significant digital predictors. Notably, intrapersonal EI and mindfulness became non-significant when technology attitudes were included. Digital centaurs predominantly used AI functionally and reported positive emotions (curiosity, pleasure, trust, gratitude) but expressed concerns about human misuse of AI. These findings position the digital centaur as an adaptive and preadaptive strategy for the technologically augmented human. This has direct implications for education, highlighting the need to foster balanced human–AI collaboration.

## 1. Introduction

Industry 4.0 is gradually confronting us with the reality of increasingly active integration between humans and technology. These changes can be described through the metaphor of the technologically augmented human. It illustrates how a person develops within a constantly evolving socio-cultural environment, where one of the main characteristics is the high speed of digital transformations. Computers, smartphones, the Internet of Things, and AI assistants together form an integrated technosystem that mediates human everyday life. These technologies act as complex cultural tools, serving as extensions and augmentations of people that ultimately become part of their very identity ([56]).

The study of socio-cultural digital artifacts as external extensions of the human being is carried out within various scientific fields. These disciplines can be viewed as different branches of externalist philosophy, such as the social externalism found in the cultural–historical approach ([65]; [11]). Researchers operationalize the processes and phenomena under study within a similar phenomenology of “the extensions of man” ([38]); “extended mind” ([10]); “augmented human intellect” ([19]); “extended self” ([4]); and “augmented human” ([20]). In this work, we draw on the socio-cognitive concept of digital socialization. Within the paradigm of cultural–historical psychology and building on the above approaches, the concept considers one of the key concepts and the main outcome of digital socialization to be the technologically augmented individual. In such an individual, cognitive, personal, and behavioral systems are fused with elements of the technosystem ([56]).

What are the current and future pathways for the technological augmentation of the human being and for our integration and fusion with technology? Drawing on both theoretical and empirical research, scholars have identified several possible trajectories for this development, which represent distinct types of the technologically augmented individual. These include digital donors, techno-conservatives, techno-isolationists, personoids, cyborgs, and digital centaurs ([57]). In this work, we focus on the digital centaur, operationalizing this type through its strategy of adaptation to a technology-saturated reality. Already being in use across certain domains of human activity, this metaphor speaks to the urgent need to enhance the quality of human intelligence. The founding figure of cyberpsychology—physicist, mathematician, and psychologist Joseph Licklider—wrote as early as 1960, in his seminal article Man–Computer Symbiosis: “The hope is that, in not too many years, human brains and computing machines will be coupled together very tightly, and that the resulting partnership will think as no human brain has ever thought and process data in a way not approached by the information-handling machines we know today” ([35]). This juncture can be considered to have arrived in November 2022 with the public accessibility of ChatGPT.

### 1.1. Digital Centaurs

A digital centaur is an individual enhanced by digital technologies and artificial intelligence, capable of solving problems more effectively than either a human or AI alone ([57]). Their adaptation strategy is characterized by a conscious and purposeful integration of artificial intelligence as a tool for augmenting personal capabilities, optimizing activities, and enhancing productivity and comfort across various domains. The core of this adaptation lies in a human-centered collaboration, where the individual actively leverages AI to solve problems more effectively than either a human or AI alone, while maintaining the integrity of the self and the dominance of the human element. The concept of enhancing human intellectual abilities predates the widespread adoption of AI. Early formulations, such as [19] ([19]) concept of “Augmenting Human Intellect,” laid the groundwork for this idea, which later evolved alongside emerging technologies like the Internet of Things and Intelligence Augmentation ([62]). Contemporary research on cognitive processes extended by digital tools rests primarily on the thesis of the extended mind ([10]) and is often situated within the broader trend known as cognitive offloading—that is, the process of “unloading” cognitive functions by using external tools ([43]). The collaborative interplay between humans and artificial intelligence is described through a variety of terms, including human–robot teams ([66]), human–machine interaction ([41]), hybrid intelligence ([2]; [14]), and collaborative intelligence ([53]). The term “centaur” is currently used in two key contexts. In cognitive science, it refers to computational models that can predict and simulate human behavior in experiments expressible in natural language ([6]). In a closely related vein, it also denotes a model of human–AI collaboration within philosophy and posthumanist ethics, known as Centaur Intelligence ([71]).

The practical embodiment of the digital centaur concept began in 1998, when the first “advanced chess” match was played. Here, the competitors were not merely chess players but human–computer pairs. As a result, chess players’ level of mastery rose dramatically, and the very approach to the game itself was transformed. Today, the strongest chess players are no longer solely human but centaurs: hybrids of humans and machines, a true symbiosis of human and artificial intelligence. In 2016, a computer defeated a human in the Chinese strategy board game Go—an achievement once thought impossible due to the game’s complexity, which defies brute-force algorithmic analysis. Yet by 2021, two of Russia’s strongest Go players, operating according to the digital centaur strategy, faced the AI system Leela Zero and emerged victorious, demonstrating the power of human–machine collaboration.

By late 2022, the digital centaur strategy had moved beyond the closed circles of professional logic-game players as everyday users gained access to ChatGPT—an “intelligent” technological extension. ChatGPT broke the world record as the fastest-growing application in terms of downloads, reaching 100 million users within just two months of its launch. For comparison, TikTok required nine months to achieve the same milestone, while YouTube took four years and one month.

To what extent is the digital centaur strategy being realized across different domains? Most research to date remains experimental in nature, yet several general conclusions can already be drawn. The integration of AI into a range of tasks—such as creative writing, report preparation, email composition, programming, and call center consulting—has been shown to improve productivity. However, the most pronounced gains are typically observed among individuals who initially possess weaker skills ([39]; [16]; [13]; [45]). For example, in an experiment with 444 professionals, ChatGPT significantly enhanced writing productivity (reducing time by 0.8 SD and improving quality by 0.4 SD) while reducing performance inequality by disproportionately benefiting lower-ability workers ([39]). Similar results were observed in a study of 758 consultants, where AI use improved performance by 43% among below-average performers, compared to a 17% increase for above-average performers ([13]). Nevertheless, human–AI collaboration enables individuals and organizations alike to acquire deeper insights, accelerate innovation, and tackle complex problems with greater efficiency across diverse sectors. As these technologies evolve, emphasis is shifting toward fostering ethical, inclusive, and human-centered collaboration—one that strengthens, rather than diminishes, collaborative intelligence ([73]).

### 1.2. Human Intelligence and Mindfulness

The ability of a digital centaur to harness artificial intelligence, integrate with it from a human-centered perspective, and counterbalance it when needed depends entirely on its foundation of strong human intelligence. This encompasses the full spectrum of “cold” and “hot” forms, such as academic, social ([31]), emotional ([37]), practical ([61]), cultural ([18]), and personal ([36]) intelligence. We argue that the principal meta-intellectual regulatory component capable of integrating the complex identity of the digital centaur may be personal intelligence. Personal intelligence enables the creation of viable strategies for identity formation, allowing individuals to both integrate and differentiate themselves. This capacity develops through cultivating individuality, agency, self-reflection, mindfulness, metacognition, and the ability to evaluate both other people and digital entities—the identity that allows the human element in people to remain dominant over the digital. In this interpretation, personal intelligence remains understudied, though research into emotional intelligence and mindfulness in the context of human–AI collaboration is advancing. While technical and operational efficiencies of AI are widely discussed, the role of human EI in enhancing this collaboration remains underexplored. Research demonstrates the strategic importance of EI in fostering effective human–AI synergy and proposes that EI-driven human resource management practices can significantly improve organizational adaptability and performance ([28]; [23]). Another study found that workplace mindfulness played a critical role in mitigating the negative impact of job insecurity on technology-related anxiety. This insecurity specifically stemmed from the integration of human–AI collaboration ([67]). Accordingly, we hypothesize that a preference for the digital centaur as an adaptation strategy of the augmented individual will be determined by emotional intelligence (H1) and mindfulness (H2).

### 1.3. Digital Predictors: User Activity, Digital Competence, and Attitudes Toward Technology

The digital centaur is, by definition, intrinsically linked to technology use. In the digital realm, screen time and user activity serve as indicators of access to technological extensions and are important determinants of well-being ([64]). At the same time, hyperconnectivity has become the norm of digital everyday life. Hyperconnectivity is conceptualized as a quantitative aspect of modern life, characterized by an environment saturated with digital devices, high user activity, and maximal screen time. These factors, in turn, lead to qualitative changes in daily functioning ([8]; [42]; [56]). It can be hypothesized that digital centaurs are likely to exhibit high levels of screen time, even to the point of hyperconnectivity (H3).

In the context of constant and rapid technological change, accompanied by rising technostress, attitudes toward technology may play a significant role in an individual’s resilience and successful adaptation to the modern world. The cognitive dimension of such attitudes is often positioned along two poles: technooptimism and technopessimism. Technooptimism reflects a worldview and life stance in which technological achievements and scientific–technical progress are assigned primary importance in addressing social problems ([33]; [51]). Technopessimism, by contrast, relates to the belief that technological progress impedes societies’ well-being and that its benefits are less than its harm ([33]; [48]). The emotional–behavioral dimension of attitudes toward technology is reflected in the phenomena of technophilia and technophobia. Technophobia is understood as an internal resistance that arises when people think or speak about new technology; it encompasses fear or anxiety related to its use, as well as hostile or aggressive attitudes toward it ([7]). Technophilia, conversely, denotes a positive disposition toward most technologies, enjoyment derived from using new ones, and a readiness to gain experience in their application ([3]; [40]). Researchers also identify technorationalism—the conscious and deliberate use of technology ([59]). We may hypothesize that digital centaurs are more likely to exhibit technophilia and technorationalism than technopessimism and technophobia (H4).

Another critical attribute of the digital centaur is digital competence—the ability to act effectively and safely in the digital environment, to use complex digital tools, and to critically evaluate the risks associated with them ([17]; [50]; [60]; [63]). Digital competence serves as a predictor of a preference for the digital centaur as an adaptation strategy of technologically augmented individuals (H5).

### 1.4. Relationship with Artificial Intelligence

In understanding the digital centaur, the individual’s relationship with their “intelligent extension” in the form of artificial intelligence is of particular significance. Based on existing research into human–AI interaction, three principal dimensions can be distinguished: AI usage practices (as a companion or as a tool), acceptance of AI across different domains, and fears and concerns associated with AI adoption.

Two areas of research have provided insights into the interaction between humans and AI: functional use and relational use. The first approach views technologies mainly as tools designed to carry out specific functions, emphasizing the necessity of their adoption and use to fulfill these practical purposes. The key theoretical frameworks within this perspective are the diffusion of innovations theory ([52]), domestication theory ([55]), and the more contemporary six-phase acceptance model for interactive technologies ([12]). In contrast, the second area posits that individuals perceive technologies as social entities, fostering personal connections with them. This perspective is largely informed by the computers as social actors (CASA) framework ([49]), which has inspired a considerable body of research ([21]). The longitudinal study examined both types of use ([69]). The results confirmed that functional and relational uses of technology mutually reinforced each other over time. Specifically, relational use enhanced future functional use, while self-disclosure strengthened relational use. Interestingly, functional use did not lead to increased relational use; rather, longitudinal mediation analysis indicated that it actually decreased relational use due to insufficient self-disclosure. An international study of AI use practices among students has shown that AI usage and positive AI attitudes significantly predict interest in AI, which, in turn, and together with AI literacy, enhance AI self-efficacy ([5]). The study also identified three groups of students: “AI Advocates,” “Cautious Critics,” and “Pragmatic Observers,” each exhibiting unique patterns of AI-related cognitive, affective, and behavioral traits.

The research field is increasingly enriched by studies demonstrating the ambivalence of attitudes toward AI: fears regarding artificial intelligence coexist with expectations of its benefits and its regular use ([15]; [27]; [34]). A comprehensive meta-analysis reveals that AI acceptance is driven by a multifaceted framework encompassing both AI characteristics and individual user factors ([34]). This dual-perspective approach, examining AI as both tool and agent, demonstrates that acceptance depends on multiple interdependent elements, including capability, role, anthropomorphism and context. A study of 554 Japanese people found a link between the frequency of AI use and acceptance of the technology ([27]). It may be hypothesized that digital centaurs are characterized by functional AI use, a positive emotional profile in interactions with AI, a high level of acceptance of AI in domains related to enhancing productivity and personal comfort, and fears primarily grounded in concerns over the unlawful or unethical use of AI by humans (H6).

As a type of technologically augmented individual, the digital centaur faces a number of specific risks. Given their deep integration with technology, digital centaurs continuously strive to unify their physical and digital dimensions. This ongoing process complicates the achievement of a stable identity and demands considerable self-reflective effort ([58]). A disruption of the balance between the human and technological elements within the digital centaur creates conditions for a new form of identity crisis, the consequences of which remain insufficiently understood. This underscores the importance of studying this strategy as one of the most adaptive in today’s hybrid online/offline reality.

The aim of this study is to identify the digital and personal predictors influencing adolescents’ and young adults’ preference for the digital centaur as a strategy of adaptation—one of the possible developmental trajectories of the technologically augmented individual in the era of artificial intelligence. These findings provide the first evidence for the personal and digital factors that shape young people’s preference for human–AI collaboration as their primary strategy. They also reveal the specific nature of their attitudes toward artificial intelligence within this context.

## 2. Materials and Methods

### 2.1. Participants

The study sample comprised 1,841 respondents: 649 adolescents aged 14–17 years (M = 16.3, SD = 0.9, 55.3% female), 817 youth aged 18–23 years (M = 19.8, SD = 1.7, 46.0% female), and 375 young adults aged 24–39 years (M = 31.0, SD = 5.2, 22.2% female). The sample comprised 17.3% secondary school students, 24.7% college students, 34% university students, and 24% employed individuals. In terms of financial status, 39.1% of participants described their income level as high, while 40.7% reported a moderate financial situation, noting that they could not afford to purchase property or a car. Additionally, 20.2% indicated that they were experiencing financial difficulties. Respondents were drawn from major urban centers across five regions of Russia, including Moscow (32.2%), Saint Petersburg (14.9%), Tyumen (14.7%), Rostov-on-Don (19.2%), and Makhachkala (19.1%).

### 2.2. Data Collection

Participants were recruited through an online survey conducted via Google Forms between autumn 2024 and winter 2025. The recruitment was carried out within a research network comprising universities, schools, and colleges. Prior to completing the questionnaire, all the participants were provided with information about the study and gave informed consent.

### 2.3. Materials and Procedure

#### 2.3.1. Socio-Demographic Questionnaire

The methodological toolkit of the study included a socio-demographic questionnaire comprising questions on gender, age, place of residence, type of employment, level of education, and socio-economic status, operationalized through family income.

#### 2.3.2. Digital Centaur

To assess participants’ preferences for interaction strategies with digital technologies and AI in the framework of the “digital centaur” concept, the vignette method was employed. The vignette is a description of a certain social situation or human experience that is assessed by a respondent based on a number of parameters ([1]). Respondents were asked to read a vignette and answer questions. The vignette was presented in two versions, reflecting both female and male gender identities. The vignette included parameters of the digital centaur, such as technooptimism, the importance of technological literacy, and the use of technology and AI to achieve success, optimize activities, and enhance productivity and comfort across various domains: “Milana (Max) believes that success in the modern world is impossible without obtaining technological literacy. She (He) actively explores innovations in the digital sphere, not only to make her (his) life more convenient but also to achieve success across various domains. Artificial intelligence helps her (him) optimize her (his) professional activities, search for information, create content, and translate texts. Milana’s (Max’s) key advantage lies in her (his) ability to enhance her capabilities through the use of technology. As a result, she (he) completes tasks more efficiently and with higher quality, which contributes to her (his) career advancement. She (He) also integrates digital tools into her (his) everyday life—from planning meals to managing a smart home—allowing her (him) to save time and devote more attention to her (his) personal interests and loved ones”.

The case was followed by these questions: “How much do you like this person?”, “How similar are you to this person?”, “How willing would you be to regularly communicate with someone who lives this kind of lifestyle?”, and “Would you like to be like this person in 10 years?” Responses were rated on a 5-point Likert scale ranging from 1 (“strongly disagree”) to 5 (“strongly agree”). The Digital Centaur Scale was constructed by calculating the mean score across the first three items (Cronbach’s α = 0.80, M = 3.30, SD = 0.96). The Digital Centaur Scale measures the overall preference for this adaptation strategy. In the analysis of the results, three indicators were used: (1) The Digital Centaur Scale, (2) Present Self-Identification (“How similar are you to this person?”), and (3) Future Self-Identification (“Would you like to be like this person in 10 years?”).

#### 2.3.3. Trait Mindfulness and Emotional Intelligence

Respondents completed the Mindful Attention Awareness Scale (MAAS) ([9]), using the Russian-language adaptation consisting of 15 items forming a single scale of mindfulness ([72]).

Emotional intelligence was measured using the Brief Version of the Emotional Intelligence Test ([44]), which consists of 8 items, with two items assessing each of the following scales: Understanding of One’s Own Emotions, Management of One’s Own Emotions, Understanding of Others’ Emotions, and Management of Others’ Emotions. These four scales are further grouped into two subscales: Intrapersonal and Interpersonal Emotional Intelligence.

#### 2.3.4. Internet Use, Attitudes Toward Technology, and Digital Competence

Participants were asked to self-assess their daily internet usage. This was carried out via the question, “On average, how much time do you spend on the internet per day?” The response scale ranged from “Up to 1 h” to “12 h or more,” with 1 h intervals.

Digital competence was assessed using the Screening Version of the Digital Competence Index, which includes 16 items comprising a single scale of digital competence. The scale evaluates knowledge, skills, motivation, and responsibility/safety across four domains: content-related activities, communication, consumption, and the technosphere ([60]).

Attitudes toward technology were assessed with the Technology Attitudes Questionnaire, which includes 20 items distributed across four subscales: openness and enthusiasm for using technology (“Technophilia”), mindful and rational use of innovative technology (“Technorationalism”), difficulties in mastering and using technology (“Technophobia”), and a critical view of the social risks of technology (“Technopessimism”). The questionnaire underwent validation on a Russian-speaking sample ([59]). The study included 808 participants, of whom 448 were parents of adolescents aged 14–17 years (M = 40 years, SD = 5.9, 77% female) and 360 were adolescents aged 14–17 years (M = 15.4 years, SD = 1.07, 50% female). The questionnaire’s structure was refined based on the results of exploratory and confirmatory factor analyses (CFA = 0.9, RMSEA = 0.07, SRMR = 0.06), and all scales demonstrated good internal consistency (Cronbach’s α = 0.66–0.88). The results reported by the questionnaire’s authors indicate its reliability and support its use for research purposes. The scales demonstrated good reliability in our sample, with all Cronbach’s alpha coefficients falling within an acceptable to good range (0.78–0.91).

#### 2.3.5. AI Use and Attitudes Toward AI

As part of the study, a set of questions was developed to assess participants’ use of and attitudes toward artificial intelligence (AI).

Participants were asked to indicate how frequently they use AI-powered chatbots for various purposes using the question, “How often do you use AI chatbots for the following purposes?” The listed purposes included completing academic tasks, solving work-related problems, entertainment, gaming and leisure, obtaining information, functioning as a personal assistant, psychological support, social interaction, receiving relationship advice, and developing specific skills and abilities. Participants were asked to rate each item on a 5-point Likert scale ranging from 1—“never” to 5—“constantly”.

Participants were also asked about the emotions they experience in relation to AI with the question, “What do you usually feel toward AI-based chatbots when interacting with them?” The listed emotional responses included interest, irritation, pleasure, anxiety, gratitude, envy, trust, and a sense of inferiority. Each item was rated on a 5-point Likert scale, where 1 indicated “never feel this emotion,” and 5 indicated “constantly feel this emotion”.

To evaluate participants’ acceptance of AI in various societal roles, they were asked: “Artificial intelligence and robotics are rapidly evolving. Today, AI is already capable of performing complex tasks, monitoring the environment, expressing emotions, and engaging in conversations indistinguishable from human interaction. To what extent would you be willing to accept AI in the future in the following roles?” The listed roles included a work colleague, a friend, a school or university teacher, a household assistant, a psychologist, a romantic partner, a physician, a judge, a nanny, a city mayor, a president, and a police officer. Each role was evaluated on a 5-point Likert scale, from 1—“completely unwilling” to 5—“completely willing”.

To assess fears related to AI and the future of this technology, participants were asked, “To what extent are you concerned about the following?” The statements included: AI will lead to the disappearance of professions and transform the labor market; AI will compete with humans as friends and romantic partners; AI will be used for the benefit of certain individuals, groups, or organizations; AI will be used to commit various crimes; humans will make critical decisions based on AI and may be mistaken; AI will go out of human control and start governing people; AI will make humans lazy and prevent their development; AI will destroy humanity; people will begin to worship AI and turn it into a religion; AI will be used to enhance government control over citizens’ lives; and AI will outperform me in everything, making me feel worthless. Participants rated each concern on a 5-point Likert scale, from 1—“not concerned at all” to 5—“very concerned”.

### 2.4. Data Analysis

Data analysis was conducted using IBM SPSS Statistics 22.0 and Jamovi 2.4.8, employing Pearson correlation coefficients, ANOVA, and hierarchical linear regression analysis.

## 3. Results

### 3.1. Digital Centaurs: Age, Gender and Financial Status

Age group differences were found among adolescents aged 14–17 years, youth aged 18–23 years, and young adults aged 24–39 years on the Digital Centaur scale (F = 20.6, df = 2, *p* < 0.001), Present Self-Identification with Digital Centaur (F = 7.7, df = 2, *p* < 0.001), and Future Self-Identification with Digital Centaur (F = 14.2, df = 2, *p* < 0.001). Youth aged 18–23 years demonstrated the highest scores across all indicators. The results are presented in Figure 1. No significant gender differences were found.

Significant differences in financial status (high, moderate, low) were found on the Digital Centaur scale (F = 4.6, df = 2, *p* < 0.01) and Future Self-Identification with Digital Centaur (F = 7.6, df = 2, *p* < 0.001). However, no significant differences were observed for Present Self-Identification with the Digital Centaur. These results are presented in Figure 2.

To calculate the percentage distribution of Digital Centaurs based on three indicators among adolescents, youth and young adults, participants were categorized into three groups: High Digital Centaur Group (scores ≥ 3.6 and ≤5.0), Moderate Digital Centaur Group (scores ≥ 2.5 and ≤3.5), and Low Digital Centaur Group (scores ≥ 1.0 and ≤2.4). The grouping was based on a ±0.5 range from the scale’s mean value of 3. At present, 27.3% of respondents identify themselves as Digital Centaurs, while 41.3% would like to become one in the next 10 years. The number of respondents who are uncertain or do not want to become Digital Centaurs declines over a 10-year time horizon. The results are presented in Figure 3. 

### 3.2. Predictors of Digital Centaur Strategy Preference

The results presented in Table 1 indicate that the Digital Centaur Scale is associated with mindfulness, intrapersonal and interpersonal emotional intelligence, as well as with digital competence, daily internet usage, technophilia, technorationalism, and weakly with technopessimism, but not with technophobia.

The results of the first stage of hierarchical regression analysis showed that mindfulness, intrapersonal and interpersonal emotional intelligence, digital competence, and user activity were significant predictors of the Digital Centaur preference (R^2^ = 0.141, *p* < 0.001). When technophilia, technopessimism, and technorationalism were added to the model at the second stage, mindfulness and intrapersonal emotional intelligence lost their significance, and technorationalism also turned out to be non-significant (R^2^ = 0.252, *p* < 0.001) (Table 2).

To clarify the mediating role of Technophilia in the preference for a Digital Centaur strategy, we performed a bootstrapped mediation analysis with 5000 resamples. The results indicate that Technophilia is a significant mediator for the effects of both Technorationality and Mindfulness on the Digital Centaur Scale. Specifically, for these two predictors, only the indirect effects through Technophilia were statistically significant, while their direct effects were not. In contrast, Digital Competency exhibited a dual influence on the Digital Centaur Scale, demonstrating both a significant direct effect and a significant indirect effect mediated through Technophilia. The complete results and statistics are presented in Figure 4 and Table 3.

### 3.3. Attitudes of Digital Centaurs Toward AI

The results presented in Table 4 show that Digital Centaurs primarily use AI for work-related and academic tasks, as well as for developing skills, but not for personal communication or leisure. Additional analysis was conducted on the frequency of AI use within the High Digital Centaur Group (based on the Digital Centaur Scale). In this group, 59.6% use AI frequently or constantly for academic tasks, 46.8% for work tasks, 26.9% for skill and ability development, and 31.6% for entertainment, gaming, and leisure.

Digital Centaurs predominantly experience positive emotions toward AI, including curiosity, pleasure from interaction, and gratitude toward the intelligent digital assistant, whom they also tend to trust. At the same time, they are less likely to experience negative emotions such as anxiety, envy, or sense of inferiority (Table 5). 

Digital Centaurs are generally willing to accept AI primarily in professional roles, such as colleagues, teachers, household assistants, doctors, and judges, and even in the more intimate role of a friend. However, they are reluctant to see AI as a nanny for their child or as a romantic partner. They also express doubts about AI’s ability to hold high-level leadership positions, such as a mayor or president (Table 6).

Digital Centaurs are primarily concerned that humans might misuse AI technology, leading to new risks such as increased crime, job loss, enhanced governmental control, or even stagnation of human development due to overreliance on AI. They do not believe that AI can replace humans, especially in personal relationships, such as in the role of a friend or romantic partner. Additionally, Digital Centaurs generally do not fear that AI will surpass them in all areas (Table 7). 

## 4. Discussion

The findings indicate that the digital centaur, conceptualized as an adaptive strategy for individuals in an increasingly complex digital world, is becoming increasingly prevalent among young people. At present, more than a quarter of respondents identify as pronounced digital centaurs, while a larger share, nearly half, prefer this strategy of human–technology interaction and desire to adopt it in the future. The prevalence of the digital centaur strategy in our sample aligns with data on ChatGPT usage among Americans for work (28%), learning (26%), and entertainment (22%) ([46]). The digital centaur is optimistic about technological progress, aware of the importance of technological literacy, and adept at using technology and AI to achieve success, optimize activities, and enhance productivity and comfort across various domains. This strategy is gradually securing an important place not only in the discourse of scholars, policymakers, and business leaders, advancing human-centered human–AI collaboration ([22]; [53]), but also in the mindset of everyday users. This trend underscores both the readiness of a substantial segment of society for such transformations and the need for targeted efforts to foster the constructive development of this strategy, particularly among youth. According to our data, the most active practitioners and those most prepared to adopt the digital centaur strategy are young people aged 18–23—primarily university students. For children and adolescents, the primary developmental task is to master their own cognitive abilities, which should serve as the foundation for future effective collaboration with AI rather than be replaced by it, whereas young adults are already in a position to actively employ AI as digital centaurs. Research confirms that this age group is indeed emerging as the most active users of AI at present: 46% of younger adults (18–29) report using AI weekly ([25]). 

Intrapersonal and interpersonal emotional intelligence, along with mindfulness, emerge as predictors of a preference for the digital centaur strategy, supporting H1 and H2. One of the risks for digital centaurs lies in the disruption of balance between the human and the technological, particularly in the context of interpersonal relationships, loss of control over intelligent extensions, and loss of integrity ([57]; [58]). It can be assumed that these personal characteristics are closely linked to self-regulation and may act as protective factors against such risks. One of the few empirical studies shows that increased workplace mindfulness mitigates the negative impact of human–AI collaboration job insecurity on tech-learning anxiety and well-being among Chinese employees who work daily with AI ([67]). This can be correlated with our research findings that mindfulness promotes a balanced use of AI within the digital centaur strategy. 

In most works, the focus shifts to how AI itself should exhibit traits associated with emotional intelligence to improve human–AI collaboration ([28]; [23]). Our results, which revealed the role of user emotional intelligence as a predictor of preference for the digital centaur, are strongly corroborated by research focused on the design of AI systems themselves. As demonstrated by the systematic review of [32] ([32]), effective human–machine collaboration requires AI to exhibit qualities associated with high emotional intelligence: empathy, rapport building, and trustworthiness. Numerous empirical findings suggest that such “socially and emotionally competent” AI significantly enhances user adherence to recommendations (by 20%), reduces anxiety (by 22%), and strengthens trust (by 40%). Therefore, it can be concluded that the successful implementation of the digital centaur strategy is based on a reciprocal process: it is determined not only by the individual’s personal characteristics but also by the AI agent’s ability to adequately respond to their emotional and social needs. The present findings contribute to the still limited body of research on the role of mindfulness and emotional intelligence in human–AI interaction.

User activity contributes to the preference for the digital centaur, confirming H3. In this way, the digital centaur fulfills its need for access to its digital extensions and, to some extent, ensures its own well-being—something that is becoming increasingly normative within the framework of digital everyday life for society as a whole ([64]). These findings are consistent with international research indicating that frequent internet users are more likely than others to report feeling excited about the increasing use of AI in daily life ([47]). At the same time, although the relationship with user activity is significant, it is relatively weak. This suggests that a digital centaur’s hyperconnectivity is defined less by actual screen time and more by the constant availability of digital tools and a highly saturated technological environment. ([8]; [42]; [56]). 

The choice of the digital centaur preference is primarily associated with technophilia and technorationalism and only weakly with technopessimism, confirming H4. This is supported by a meta-analysis confirming the significant role of attitudes toward technology in shaping behavioral choices ([30]). The data obtained reveal a complex picture of the digital centaur’s relationship with technology. The digital centaur maintains positive, open, and enjoyable engagement with technology, yet approaches it with mindfulness and functional purpose. This approach also involves an acknowledgment of technology’s potential risks, limitations, and more pessimistic scenarios. This suggests that, at the current stage of human–machine symbiosis, this strategy of a technologically augmented individual is relatively balanced. Furthermore, mediation analysis revealed that technorationalism is significant only through its association with technophilia. Thus, a general positive attitude toward technology is a stronger predisposing factor for preferring the digital centaur strategy. These results correspond with the AI Device Use Acceptance Model, which demonstrates the role of hedonic motivation ([26]). Hedonic motivation refers to the perceived pleasure derived from using an AI device. The model’s findings indicate that hedonic motivation is positively related to AI performance expectancy. Digital competence serves as a key foundation for this balance and a significant predictor of the digital centaur preference (H5). It provides the motivation, knowledge, and skills necessary for the effective, responsible, and safe use of digital technologies across various domains. This is also consistent with research showing a positive association between students’ digital competence and their use of AI ([29]). Digital competence provides the basis for a technologically augmented human to self-regulate their extensions. At the same time, a positive attitude toward technology, when combined with higher levels of digital competence and user activity, appears to be more influential in determining the choice of the digital centaur strategy than mindfulness or intrapersonal emotional intelligence. At this stage, digital predictors play a more significant role, sharpening the question of the need for deliberate efforts to preserve the human element in individuals augmented by technology.

The findings regarding relationships with AI confirm H6. For digital centaurs, AI use is characterized more by a functional than relational usage for work/study, self-development, entertainment, and leisure, thus framing AI primarily as a tool ([52]; [55]; [12]). However, when emotions are analyzed, the Computers as Social Actors (CASA) framework ([49]; [21]) becomes equally relevant. Alongside emotions such as contentment (pleasure) and amusement (curiosity), the digital centaur also experiences connection emotions (gratitude, trust) that are typically associated with interpersonal relationships. As in other studies, this demonstrates that functional and relational use in fact complement each other ([69]). The qualitative study of emotions toward AI in the workplace identified a broad emotional spectrum ([24]), but unlike those findings, digital centaurs tend not to exhibit frustration emotions, such as envy, a sense of inferiority, or anxiety. The results highlight the specificity of AI-induced emotions among digital centaurs and expand the current research, which has largely been conducted in the context of work and education (e.g., [24]; [54]; [68]; [70]).

As previous research has shown, AI acceptance and AI-related fears are complementary categories ([15]; [27]; [34]). International data further reveal that public sentiment is often mixed; for instance, 42% of respondents report feeling equally concerned and excited about AI’s growing presence in daily life ([47]). In our study, AI acceptance was assessed across various social roles. Digital centaurs are already prepared to accept AI in many professional roles—more often as a household assistant or workplace colleague, less often as a physician or teacher. However, two categories currently demonstrate low levels of acceptance. The first includes social roles that imply close personal relationships, such as a romantic partner or a nanny who would be a significant figure in a child’s life. The second encompasses domains associated with leadership and high-stakes global decision-making, where the consequences and cost of errors are extremely high, such as the roles of a city mayor or a country’s president. A large-scale international study on fears regarding AI in six professions (doctors, judges, managers, care workers, religious workers, and journalists) found results that varied substantially across cultures ([15]). For example, in Russia, AI was perceived as more acceptable among religious workers, journalists, care workers, and managers than among doctors or judges, a finding that partially aligns with our own data. Fears toward AI among digital centaurs are not strongly pronounced, which reflects their generally critical stance toward technology. Their concerns are focused less on AI itself and more on its potential for misuse by humans, such as for selfish or criminal purposes that restrict human freedom or suppress individuals. It can be assumed that individuals who prefer the digital centaur strategy exhibit a higher degree of AI acceptance and less pronounced fears than society at large.

The present study has several limitations that should be considered when interpreting the results and planning future research. First, the data collection relied exclusively on self-report measures, which may not fully capture actual behavioral patterns and the true prevalence of the digital centaur strategy. Furthermore, the methodology assessed the preference for the digital centaur as a strategy of the technologically augmented individual, rather than directly measuring the corresponding collaborative behavior with AI in real-world settings. To obtain a more objective picture, future studies should supplement questionnaires with behavioral data, such as log analysis of AI interactions or experimental methods.

Second, the study sample was drawn from urban centers in Russia, which limits the generalizability of the findings to rural populations who may have different levels of access to digital technologies, digital competence, and attitudes toward technology.

Third, the mindfulness assessment tool used (MAAS) operationalizes mindfulness primarily in terms of present-moment attention and awareness of internal and external experiences. This leaves out other important facets of mindfulness, such as acceptance or curiosity, which could also play a significant role in shaping a preference for the digital centaur strategy.

The phenomenon of the digital centaur is only beginning to be empirically studied in psychology, leaving room for a broad spectrum of future research. To gain a deeper understanding of the nature of digital centaur activity, it is advisable to employ mixed-methods research designs, combining quantitative surveys with qualitative interviews, focus groups, and experimental paradigms. This would help uncover unique patterns of human–AI collaboration and the meanings users ascribe to this partnership.

A particularly important direction is a theoretical and empirical investigation of personal intelligence as the core meta-intelligence of the technologically augmented human. It is crucial to conceptualize its structure and correlate its components with the successful implementation of the digital centaur strategy. Future research should aim to identify which specific facets of personal intelligence—such as self-reflection, agency, identity integration, metacognition, and the capacity for evaluating both humans and digital entities—are most critical for fostering a positive and adaptive variant of the digital centaur. Understanding this would make it possible to predict the risk of losing integrity and control over one’s digital augmentations and, conversely, to develop strategies for the constructive evolution of this type of technologically augmented individual, enhancing their adaptation and well-being in the AI era. Empirical validation of these components will be key for creating targeted educational interventions.

## 5. Conclusions

This study provides the first empirical identification of a set of digital and personal predictors for the preference for the digital centaur strategy among adolescents and young adults. This strategy represents one of the key developmental trajectories for the technologically augmented individual in the era of AI. The findings demonstrate the growing prevalence and appeal of the digital centaur strategy for a significant portion of young people, both today and to an even greater extent in the future. This trend reflects a desire among adolescents and young adults to engage in preadaptive behavior. Such behavior is partially realized in the present but is primarily perceived as essential for navigating the future, especially amid the rapid evolution of AI. In this sense, adopting the digital centaur strategy can itself be seen as a predictor of preadaptation. 

The study’s results have direct practical implications, particularly for the field of education. The identified preadaptive potential of the digital centaur strategy among young people highlights the need for its targeted cultivation within educational systems. Curricula should be designed not only to develop digital competence and technological literacy but also to deliberately foster the meta-intellectual level. These qualities enable the self-management of complex digital augmentations, ensuring the integrity of the self and the dominance of the human element in human–machine collaboration.

In the context of human–AI collaboration, this work underscores the importance of considering not only technological but also psychological factors when designing effective symbiotic systems. Understanding what drives digital centaurs—functionality, positive technology attitudes (technophilia), and digital competence—informs the development of interfaces and AI solutions that augment human capabilities rather than replace them. Furthermore, the identified skepticism of digital centaurs towards using AI in high-stakes social roles (e.g., nanny, political leader) and their concerns about the unethical use of AI by humans point to the necessity of advancing AI ethics and digital citizenship, making these topics critical components of future educational programs aimed at fostering balanced and human-centered collaboration. An important implication is the need to foster interest in collaborative AI use among individuals with lower socio-economic status. As our data show, they are less engaged with this technology and risk being unable to reap its corresponding benefits and improve their well-being, potentially widening existing social disparities.

## Figures and Tables

**Figure 1 behavsci-15-01487-f001:**
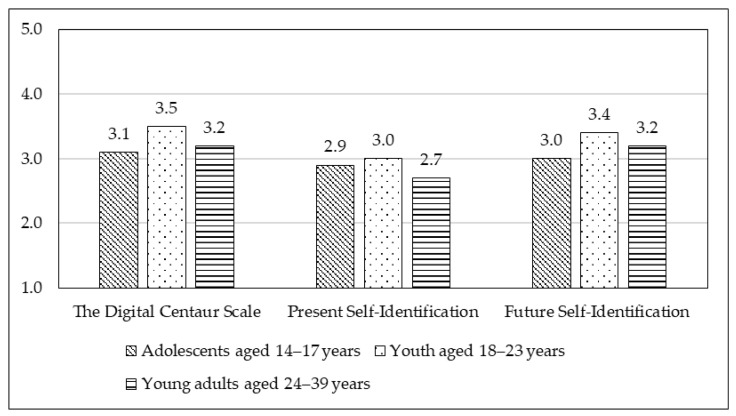
Differences in mean scores across three measures: the Digital Centaur Scale, Present Self-Identification with the Digital Centaur, and Future Self-Identification with the Digital Centaur among three age groups: adolescents aged 14–17 years, youth aged 18–23 years, and young adults aged 24–39 years.

**Figure 2 behavsci-15-01487-f002:**
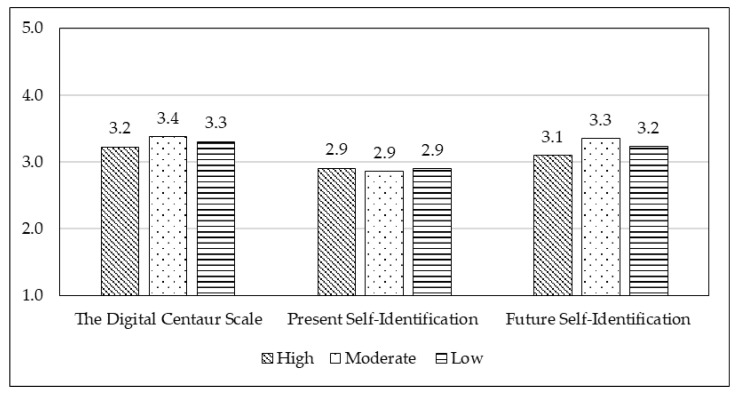
Differences in mean scores across three measures: the Digital Centaur Scale, Present Self-Identification with the Digital Centaur, and Future Self-Identification with the Digital Centaur among three groups: high, moderate and low financial status.

**Figure 3 behavsci-15-01487-f003:**
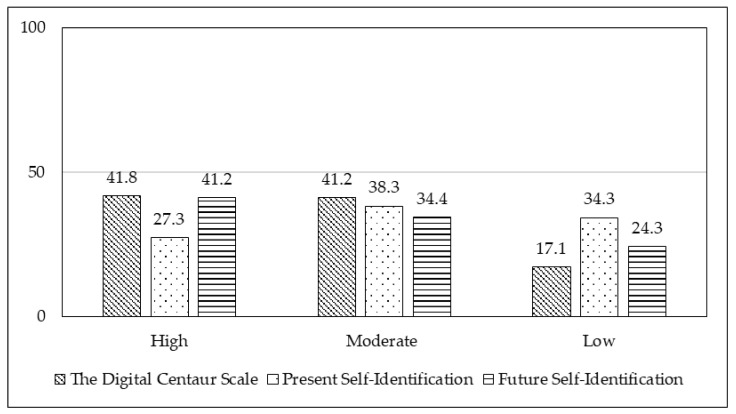
Percentage of participants in the total sample across the Digital Centaur Scale, Present Self-Identification with the Digital Centaur, and Future Self-Identification with the Digital Centaur groups, divided into High Group (scores ≥ 3.6 and ≤5.0), Moderate Group (scores ≥ 2.5 and ≤3.5), and Low Group (scores ≥ 1.0 and ≤2.4).

**Figure 4 behavsci-15-01487-f004:**
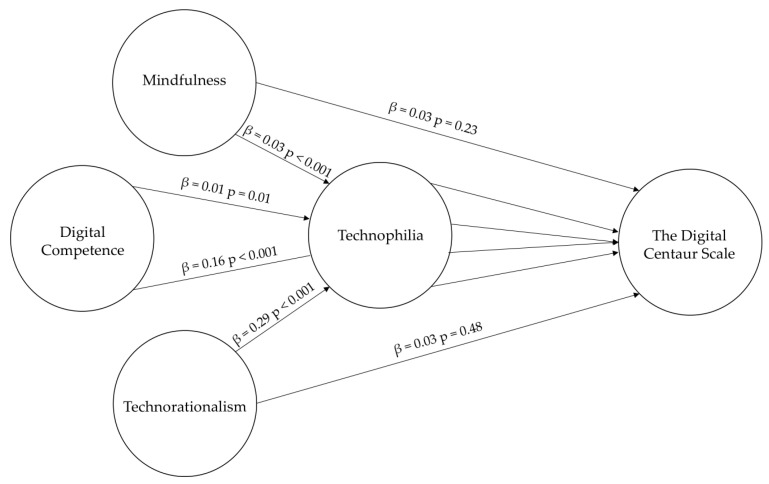
Model illustrating the direct and indirect effects of Technophilia, based on a bootstrapped mediation analysis with 5000 resamples. The figure displays the coefficients β and their *p*-values.

**Table 1 behavsci-15-01487-t001:** Pearson correlations between the Digital Centaur Scale and digital/personal indicators.

Scale	The Digital Centaur Scale
Mindfulness	0.225 **
Interpersonal Emotional Intelligence	0.167 **
Intrapersonal Emotional Intelligence	0.212 **
Digital competence	0.259 **
Technopessimism	0.119 **
Technophilia	0.450 **
Technorationalism	0.387 **
Technophobia	0.030
Daily Internet Usage	0.134 **

Note: **—*p* < 0.001.

**Table 2 behavsci-15-01487-t002:** Hierarchical Regression Model for Predictors of the Digital Centaur Scale.

Model	Predictors	Standardized Coefficient β	SE Coefficient B	*p*-Value
Step 1	Daily Internet Usage	0.101	0.008	0.000
	Interpersonal Emotional Intelligence	0.054	0.009	0.028
	Intrapersonal Emotional Intelligence	0.128	0.010	0.000
	Mindfulness	0.169	0.020	0.000
	Digital Competence	0.202	0.001	0.000
Step 2	Daily Internet Usage	0.054	0.007	0.009
	Interpersonal Emotional Intelligence	0.015	0.009	0.529
	Intrapersonal Emotional Intelligence	0.070	0.009	0.002
	Mindfulness	0.041	0.021	0.079
	Digital Competence	0.147	0.001	0.000
	Technopessimism	−0.111	0.024	0.000
	Technophilia	0.386	0.038	0.000
	Technorationalism	0.047	0.037	0.239

**Table 3 behavsci-15-01487-t003:** Direct and Indirect Effects from the Bootstrapped Mediation Model (N = 5000).

Type	Effect	SE	β	95% C.I.	*p*-Value
Indirect	Mindfulness ⇒ Technophilia ⇒ The Digital Centaur Scale	0.006	0.033	0.018–0.044	<0.001
	Digital competence ⇒ Technophilia ⇒ The Digital Centaur Scale	0.000	0.013	0.000–0.001	0.012
	Technorationalism ⇒ Technophilia ⇒ The Digital Centaur Scale	0.027	0.289	0.217–0.322	<0.001
Direct	Mindfulness ⇒ The Digital Centaur Scale	0.021	0.028	−0.022–0.073	0.225
	Digital competence ⇒ The Digital Centaur Scale	0.001	0.162	0.005–0.009	<0.001
	Technorationalism ⇒ The Digital Centaur Scale	0.035	0.026	−0.042–0.090	0.480

**Table 4 behavsci-15-01487-t004:** Digital Centaurs AI use (Pearson correlations).

Scale	The Digital Centaur Scale
Completing academic tasks	0.181 **
Resolving work-related tasks	0.178 **
Entertainment, gaming and leisure	0.094 **
Obtaining information	0.087
Functioning as a personal assistant	0.048
Psychological support	−0.055
Social interaction	−0.073
Receiving relationship advice	0.015
Developing specific skills and abilities	0.132 **

Note: **—*p* < 0.001.

**Table 5 behavsci-15-01487-t005:** AI-induced emotions (Pearson correlations).

Scale	The Digital Centaur Scale
Curiosity	0.356 **
Irritation	−0.047
Pleasure	0.221 **
Anxiety	−0.099 **
Gratitude	0.188 **
Envy	−0.093 **
Trust	0.178 **
Sense of inferiority	−0.096 **

Note: **—*p* < 0.001.

**Table 6 behavsci-15-01487-t006:** Social roles where Digital Centaurs are open to integrating AI (Pearson correlations).

Scale	The Digital Centaur Scale
A work colleague	0.237 **
A friend	0.070 **
A school or university teacher	0.168 **
A household assistant	0.325 **
A psychologist	0.065 **
A romantic partner	−0.034
A physician	0.156 **
A judge	0.112 **
A nanny	0.052
A city mayor	0.050
A president	0.006
A police officer	0.087 **

Note: **—*p* < 0.001.

**Table 7 behavsci-15-01487-t007:** Digital Centaurs’ fears and concerns regarding AI (Pearson correlations).

Scale	The Digital Centaur Scale
AI will lead to the disappearance of professions and transform the labor market	0.066 **
AI will compete with humans as friends and romantic partners	−0.065 **
AI will be used for the benefit of certain individuals, groups, or organizations	0.119 **
AI will be used to commit various crimes	0.151 **
Humans will make critical decisions based on AI and may be mistaken	0.120 **
AI will go out of human control and start governing people	0.017
AI will make humans lazy and prevent their development	0.115 **
AI will destroy humanity	−0.032
People will begin to worship AI and turn it into a religion	−0.055
AI will be used to enhance government control over citizens’ lives	0.143 **
AI will outperform me in everything, making me feel worthless	−0.074 **

Note: **—*p* < 0.001.

## Data Availability

The data presented in this study are available on request from the corresponding author.

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
