# Peer review of "The Digital Centaur as a Type of Technologically Augmented Human in the AI Era: Personal and Digital Predictors"

_behavsci, 2025, doi:10.3390/bs15111487_

Round 1
Reviewer 1 Report
Comments and Suggestions for Authors
The article explores how the concept of the "digital centaur" a technologically augmented human, can be explained through personal and digital predictors in the AI era. It is original an relevant, as it links psychological traits and digital competences to human-AI integration, an aspect less explored in the literature. It introduces the metaphor of "digital centaur" and provides a conceptual model that connects individual characteristics with digital skills, offering a novel framework for analizing human-AI augmentation.
This is an original paper with solid theory, clear methods, and well-presented results. To improve it, I suggest:
- Simplify some long sentences for better readability.
- Add a clearer comparison with previous findings.
- Expand limitations and future research with specific suggestions.
- Give more concrete examples of practical implications for education and human-AI collaboration.
Additional comments:
- They should clarify the concept's measures, use a broader sample, and apply mixed methods to improve validity.
- Conclusions are consistent and aligned with the research question, though some broader implications would benefit from more empirical support.
- The references are appropriate and up date, but including more empirical studies on AI-assisted human performance could further strengthen the paper.
- Tables an figures are clear, though they could be improved with more visual models and simplified formats to aid readability.
Comments on the Quality of English Language
The english is clear, but some sentences are too long and could be simplified to improve readability.
Author Response
We would like to express our sincere gratitude to the Reviewer for the insightful comments and valuable suggestions. We have thoroughly revised the manuscript based on this feedback and believe that it has significantly improved the quality of our work.
Comments 1: [Simplify some long sentences for better readability.]
Response 1: [We have revised the English to enhance the overall fluency and precision of the text. ]
Comments 2: [Add a clearer comparison with previous findings.]
Response 2: [ We thank the reviewer for this suggestion. To address it, we have now included a more thorough discussion that directly compares our results with key previous studies in the field. This addition can be found in the revised Discussion section.]
Comments 3: [Expand limitations and future research with specific suggestions.]
Response 3: [We have expanded the 'Limitations and Future Research' part in the Discussion section with more specific suggestions.]
Comments 4: [Give more concrete examples of practical implications for education and human-AI collaboration.]
Response 4: [We have provided more concrete examples of practical implications for education and human-AI collaboration in the Conclusion.]
Additional comments:
Comments 5: [They should clarify the concept's measures, use a broader sample, and apply mixed methods to improve validity.]
Response 5: [Additionally, the results section has been revised to provide greater clarity and detail. Specifically, an analysis of socioeconomic diversity has been added to the Results section. Furthermore, to clarify the role of Technophilia as a mediator in the relationship between Technorationality and Mindfulness, an additional bootstrapped mediation analysis was conducted (5,000 resamples). The results of this mediation analysis have been added to the manuscript.
Limitations related to the study sample were also outlined.]
Comments 6: [Conclusions are consistent and aligned with the research question, though some broader implications would benefit from more empirical support.]
Response 6: [Thank you for the comment; we have expanded the conclusion.]
Comments 7: [The references are appropriate and up date, but including more empirical studies on AI-assisted human performance could further strengthen the paper. ]
Response 7: [Following the reviewer's suggestion, we have added references to empirical work on human-AI collaboration.]
Comments 8: [Tables an figures are clear, though they could be improved with more visual models and simplified formats to aid readability ]
Response 8: [The descriptions of the figures used in the article have been revised and expanded to enhance their clarity and facilitate the interpretation of the results.]
Reviewer 2 Report
Comments and Suggestions for Authors
Thank you for this interesting paper, which taught me a lot and is highly readable. This is a very engaging area for study. I have some comments to strengthen the paper, listed below, which also identify some small mistakes.
The abstract: “adaptive and preadaptive strategies of technologically augmented human” should be “of the technologically augmented human.”
I think something has happened to the line spacing and format between lines 68 – 70. Check.
Line 70 gives a definition of digital centaur that could do with a citation.
Intro: “That time has long since arrived, but its pulse became palpably felt…” is awkward phrasing and could be more formal or less prosaic for international readers to understand.
The introduction arguably references too many theoretical traditions (extended mind, extended body, cyborgs, Industry 5.0, CASA, etc.), which risks diluting the central “digital centaur” argument. A tighter framing would improve clarity. Could make links to the work of Donna Jeanne Haraway if you were to base this more on cyborgs, but I think it's up to you, rather than a prescriptive suggestion.
Methods: “2.1. Data collection” numbering skips 2.2. Check carefully.
The heavy reliance on self-report surveys risks social desirability bias, such as attitudes toward AI. You have no triangulation with behavioural data, which might enrich the study. Also, do the authors consider the participant sample may be skewed given these are drawn exclusively through urban, rather than rural areas, where technology access may be limited. Is this deliberate? Maybe state that differences are likely to exist.
Participants section: Percentages use commas (32,2%) instead of periods (32.2%), inconsistent with APA/English norms.
Vignette text has awkward possessives: “Milana's (Max 's)” , which should be “Max’s.”
Abstract phrasing: The final sentence is awkward: “underscore digital centaur as an adaptive and preadaptive strategies…” should be “as adaptive and preadaptive strategies.”
At times “digital centaur” is described as a strategy, other times as a type of person or identity. You need consistency.
Some paragraphs contain long chains of references with little synthesis (e.g., the externalism/extended mind section). Reads more like a literature dump.
Figures 1 and 2 are included but the captions are very minimal (“Differences in Digital Centaur Scale…”). Stronger interpretive captions would improve readability.
ANOVA p-values listed as “p = 0.000.” Should be reported as “p < .001.”
Regression tables list “SE” but appear to actually report standardized beta + SE, which is unusual. This needs clarification.
I think some critique or limitations of the Mindful Attention Awareness Scale (MAAS) might be acknowledged since it is central here.
Author Response
We would like to express our sincere gratitude to the Reviewer for the insightful comments and valuable suggestions. We have thoroughly revised the manuscript based on this feedback and believe that it has significantly improved the quality of our work.
Comments 1: [The abstract: “adaptive and preadaptive strategies of technologically augmented human” should be “of the technologically augmented human.” ]
Response 1: [Thank you for your attention, we have made the correction.]
Comments 2: [I think something has happened to the line spacing and format between lines 68 – 70. Check.]
Response 2: [Thank you for your attention, we have made the correction.]
Comments 3: [Line 70 gives a definition of digital centaur that could do with a citation.]
Response 3: [Thank you for your attention, we have made the correction.]
Comments 4: [Intro: “That time has long since arrived, but its pulse became palpably felt…” is awkward phrasing and could be more formal or less prosaic for international readers to understand.]
Response 4: [Thank you for your comment, we have made the correction.]
Comments 5: [The introduction arguably references too many theoretical traditions (extended mind, extended body, cyborgs, Industry 5.0, CASA, etc.), which risks diluting the central “digital centaur” argument. A tighter framing would improve clarity. Could make links to the work of Donna Jeanne Haraway if you were to base this more on cyborgs, but I think it's up to you, rather than a prescriptive suggestion.]
Response 5: [Thank you for your insightful comments, which have been very helpful in improving the clarity of our manuscript. We have taken your feedback into consideration and have omitted the theoretical constructs that were only tangentially related to our core topic, including "extended body" and "Industry 5.0." Our continued focus is on presenting a comprehensive overview of different externalism approaches, which are still present in the text. We have retained the CASA framework as it is integral to our comparative analysis of functional and relational AI usage. We are acquainted with Donna Hathaway's work and it indeed informs our conceptualization of the cyborg strategy. However, we draw a conceptual distinction between it and the digital centaur strategy, which is why we have not included it in this paper.]
Comments 6: [Methods: “2.1. Data collection” numbering skips 2.2. Check carefully.]
Response 6: [Thank you for your attention, we have made the correction.]
Comments 7: [The heavy reliance on self-report surveys risks social desirability bias, such as attitudes toward AI. You have no triangulation with behavioural data, which might enrich the study. Also, do the authors consider the participant sample may be skewed given these are drawn exclusively through urban, rather than rural areas, where technology access may be limited. Is this deliberate? Maybe state that differences are likely to exist.]
Response 7: [Thank you very much for your valuable comments, particularly concerning the reliance on self-report surveys and the potential for social desirability bias. We also appreciate your suggestion about triangulating with behavioral data.
We wholeheartedly agree with your points and will incorporate them into the limitations and future research section of the manuscript. Our current plan includes conducting experimental studies to gather behavioral data. The rationale for our initial approach was to first collect data from a broad sample and subsequently explore these themes experimentally.
We also understand your concern about the participant sample potentially being skewed due to recruitment solely from urban areas. We will address this directly in the manuscript. While urban populations constitute approximately 75% of the general population according to 2025 data, suggesting a relatively representative sample for this demographic, we concur that it is important to state that distinct findings may emerge from rural populations.]
Comments 8: [Participants section: Percentages use commas (32,2%) instead of periods (32.2%), inconsistent with APA/English norms. ]
Response 8: [Thank you for your attention, we have made the correction.]
Comments 9: [Vignette text has awkward possessives: “Milana's (Max 's)” , which should be “Max’s.” ]
Response 9: [Thank you for your attention, we have made the correction.]
Comments 10: [Abstract phrasing: The final sentence is awkward: “underscore digital centaur as an adaptive and preadaptive strategies…” should be “as adaptive and preadaptive strategies.” ]
Response 10: [Thank you for your attention, we have made the correction.]
Comments 11: [At times “digital centaur” is described as a strategy, other times as a type of person or identity. You need consistency.]
Response 11: [We are grateful for this pertinent observation regarding the description of the digital centaur. In our study, we indeed investigate several types of the technologically augmented individual (digital donors, techno-conservatives, techno-isolationists, personoids, cyborgs, and digital centaurs (Soldatova et al., 2024)). However, as stated in the introduction, the present work specifically focuses on the digital centaur. We analyze this type through its strategy of adaptation to technological reality. We have made this more explicit in the introduction. When presenting the empirical results, we have consistently used the term "strategy" and have revised the text accordingly.
We have retained the use of "identity" in our description specifically when discussing the assessment results concerning respondents' present and future self-identification.]
Comments 12: [Some paragraphs contain long chains of references with little synthesis (e.g., the externalism/extended mind section). Reads more like a literature dump. ]
Response 12: [Thank you for pointing out the extensive chains of references in some paragraphs, specifically in the externalism/extended mind section. We understand your concern that it may read as a “literature dump”.
We acknowledge that some paragraphs contain extensive reference chains. In the externalism section, our intention was to showcase key foundational works within this research area by presenting various authors' perspectives on the phenomenon. However, we agree that the presentation of references for empirical studies could be more synthesized. We have expanded the descriptions of these empirical studies to more clearly demonstrate their relevance and connection to our work, aiming for greater synthesis. ]
Comments 13: [Figures 1 and 2 are included but the captions are very minimal (“Differences in Digital Centaur Scale…”). Stronger interpretive captions would improve readability.]
Response 13: [The descriptions of the figures used in the article have been revised and expanded to enhance their clarity and facilitate the interpretation of the results.]
Comments 14: [ANOVA p-values listed as “p = 0.000.” Should be reported as “p < .001.” ]
Response 14: [Thank you for your attention, we have made the correction.]
Comments 15: [Regression tables list “SE” but appear to actually report standardized beta + SE, which is unusual. This needs clarification. ]
Response 15: [In Table 2, which presents the results of the Hierarchical Regression Model, the SE column reports the standard error of the unstandardized coefficient B. For clarification, this has been explicitly noted in the table.]
Comments 16: [I think some critique or limitations of the Mindful Attention Awareness Scale (MAAS) might be acknowledged since it is central here. ]
Response 16: [We appreciate your insightful comment regarding the limitations of the Mindful Attention Awareness Scale (MAAS). We used the MAAS because it was the only suitable, validated method. We agree that the MAAS, while valuable, focuses on operationalizing mindfulness in terms of present-moment attention and awareness of internal and external experiences. This approach does indeed narrow the overall understanding of mindfulness. We have taken your feedback into consideration and have added this discussion to the limitations of our study.]
Additionally, the results section has been revised to provide greater clarity and detail. Specifically, an analysis of socioeconomic diversity has been added to the Results section. Furthermore, to clarify the role of Technophilia as a mediator in the relationship between Technorationality and Mindfulness, an additional bootstrapped mediation analysis was conducted (5,000 resamples). The results of this mediation analysis have been added to the manuscript.
Reviewer 3 Report
Comments and Suggestions for Authors
The study demonstrates strength in its cross-regional diversity across five Russian urban centers (Moscow, Saint Petersburg, Tyumen, etc.), enhancing ecological validity for urban youth populations. Age stratification (adolescents 14–17, youth 18–23, young adults 24–39) enables nuanced developmental analysis, revealing peak digital centaur identification among 18–23-year-olds. However, the absence of rural participants and older cohorts (40+) limits understanding of digital divides and lifespan tech adoption patterns. Socioeconomic diversity is noted (20.1% reported financial hardship), yet its impact remains unanalyzed.
1. The "digital centaur" construct is operationalized innovatively through vignettes and dual temporal identification (current/aspirational), validated by acceptable scale reliability (α=0.80). Comprehensive predictor measurement includes emotional intelligence (brief test), mindfulness (MAAS), and digital competence (DQ Index). Key gaps include unreported reliability for the techno-rationalism subscale and lack of discriminant validity testing against digital competence or technophilia.
2. Operationalization of the "digital centaur" construct via vignettes shows innovation, with acceptable scale reliability. The temporal distinction between current and aspirational identity enriches the analysis. Comprehensive measurement of predictors includes validated instruments for emotional intelligence and mindfulness. Key gaps remain in unreported reliability for techno-rationalism subscales and insufficient discriminant validity testing against related constructs like digital competence.
3. The reported non-significant predictive power of techno-rationalism (β=0.032, p=.121) raises fundamental questions about its conceptual definition. The manuscript fails to specify whether subdimensions were differentiated (e.g., instrumental rationality focusing on efficiency optimization vs. value rationality concerning ethical tradeoffs). Also why item-total correlations were omitted despite observed weak inter-scale correlations (r=.04–.07 with digital competence/technophilia)?
4. I suggest conducting Confirmatory Factor Analysis (CFA) with maximum likelihood estimation to test hypothesized factor structures, and include sample items in supplementary materials to demonstrate face validity.
5. Sole reliance on self-reported DQ Index (1–10 scale) risks illusory superiority bias, particularly given: the minimal correlation between DQ scores and actual AI usage frequency (r=.08–.11).
6. The complete mediation of digital competence (β reduction from 0.202 to 0.084) and mindfulness effects upon technophilia's introduction demands rigorous investigation. Was Sobel testing or bootstrapped mediation analysis (5,000 resamples) performed to verify indirect pathways? Potential confounding from algorithm aversion (negative experiences with AI tools) remains unaddressed.
Author Response
The authors are deeply grateful to the Reviewer for their thorough engagement with our work and thoughtful critique. The Reviewer's suggestions were exceptionally perceptive and have prompted the authors to refine the arguments and enhance the clarity of the presentation. The paper has been substantially improved as a result of this input.
We have now addressed the reviewers' comments in detail below.
Specifically, an analysis of socioeconomic diversity has been added to the Results section.
Comments 1,2,4: [1. The "digital centaur" construct is operationalized innovatively through vignettes and dual temporal identification (current/aspirational), validated by acceptable scale reliability (α=0.80). Comprehensive predictor measurement includes emotional intelligence (brief test), mindfulness (MAAS), and digital competence (DQ Index). Key gaps include unreported reliability for the techno-rationalism subscale and lack of discriminant validity testing against digital competence or technophilia. .
- Operationalization of the "digital centaur" construct via vignettes shows innovation, with acceptable scale reliability. The temporal distinction between current and aspirational identity enriches the analysis. Comprehensive measurement of predictors includes validated instruments for emotional intelligence and mindfulness. Key gaps remain in unreported reliability for techno-rationalism subscales and insufficient discriminant validity testing against related constructs like digital competence.
- I suggest conducting Confirmatory Factor Analysis (CFA) with maximum likelihood estimation to test hypothesized factor structures, and include sample items in supplementary materials to demonstrate face validity. ]
Response 1,2,4: [In the present study, a Technology Attitudes Questionnaire was employed, which underwent validation on a Russian-speaking sample, as reported by the questionnaire’s authors in 2021. The study included 808 participants, comprising 448 parents of adolescents aged 14–17 years (mean age = 40 years, SD = 5.9, 77% female) and 360 adolescents aged 14–17 years (mean age = 15.4 years, SD = 1.07, 50% female). The questionnaire’s structure was refined based on the results of exploratory and confirmatory factor analyses (CFA = 0.9, RMSEA = 0.07, SRMR = 0.06) and consisted of four scales: technophilia, technophobia, technorationalism, and technopessimism (Cronbach’s α = 0.66–0.88). All scales demonstrated good internal consistency. The results reported by the questionnaire’s authors indicate its reliability and support its use for research purposes. The questionnaire comprises only these four scales and does not include additional indicators or subscales.
The Technorationalism scale in the Technology Attitudes Questionnaire consists of four items (translated into English):
- I try to use new technological innovations if they are genuinely useful to me.
- I believe that technological innovations should be approached consciously and that one should know how to use them.
- I use new technological innovations if they can be obtained at reasonable prices.
- When deciding whether to adopt a new technology, I try to weigh all the pros and cons for myself.
All items are designed to measure the mindful and rational use of innovative technologies.
The reference is provided in the article text: Soldatova, G.U., Nestik, T.A., Rasskazova, E.I., & Dorokhov, E.A. (2021). Psychodiagnostics of technophobia and technophilia: Testing a questionnaire of attitudes towards technology for adolescents and parents. Social Psychology and Society, 12(4), 170–188. https://doi.org/10.17759/sps.2021120410 (In Russian)
The scales demonstrated good reliability in our sample, with all Cronbach's alpha coefficients falling within an acceptable to good range (0.78-0.91).
The description of the questionnaire in the Methods section has been expanded to allow any reader to familiarize themselves with the development and validation results of the Technology Attitudes Questionnaire.]
Comments 3,6: [3. The reported non-significant predictive power of techno-rationalism (β=0.032, p=.121) raises fundamental questions about its conceptual definition. The manuscript fails to specify whether subdimensions were differentiated (e.g., instrumental rationality focusing on efficiency optimization vs. value rationality concerning ethical tradeoffs). Also why item-total correlations were omitted despite observed weak inter-scale correlations (r=.04–.07 with digital competence/technophilia)?
- The complete mediation of digital competence (β reduction from 0.202 to 0.084) and mindfulness effects upon technophilia's introduction demands rigorous investigation. Was Sobel testing or bootstrapped mediation analysis (5,000 resamples) performed to verify indirect pathways? Potential confounding from algorithm aversion (negative experiences with AI tools) remains unaddressed.]
Response 3,6: [The study revealed moderate positive statistically significant correlations (not minor) of the Digital Centaur Scale with Technorationalism (r = 0.4, p < 0.001) and Digital Competence (r = 0.3, p < 0.001) (Table 1). As the study employed validated and piloted instruments and questionnaires, only the scales of these measures were analyzed, and no additional item-level analyses were conducted. The specific relationships of these variables with the Digital Centaur Scale were further examined using hierarchical regression analysis and are presented in Table 2.
We thank the Reviewer for suggesting the need for additional analysis. This suggestion allowed us to present more comprehensive results.
The minimal predictive power of technorationalism (β = 0.047, p = 0.239) at Stage 2 of the hierarchical regression analysis (Table 2) was further examined in response to the reviewer's feedback. The results indicate that technorationalism (conceptualized in this study as the mindful and rational use of innovative technologies within the Technology Attitudes Questionnaire) proves to be non-significant within a model where the overall predisposition to enjoy using technology, Technophilia (β = 0.386, p < 0.001), is a dominant factor. In contrast, both Digital Competence (encompassing knowledge, skills, motivation, and responsibility/safety) and Daily Internet Usage remained significant predictors.
Consequently, while a preference for the Digital Centaur strategy demonstrates a moderate positive association with technorationalism, this relationship is mediated by technophilia. To clarify this, a bootstrapped mediation analysis (5,000 resamples) was conducted, which confirmed the mediating role. The findings from this mediation analysis have been incorporated into the manuscript.
Future research could further clarify the nature of these relationships and the contributions of the variables by refining or expanding the model. This could involve elaborating on the technorationalism construct and employing alternative methods for assessing mindfulness, given that the MAAS scale primarily focuses on mindful attention.
As shown in Table 2, the introduction of Technophilia (β = 0.386, p < .001) in the second stage of the hierarchical regression analysis produced a consistent mediation effect. The influence of Digital Competence decreased (from β = 0.202 to β = 0.147, though remaining significant at p < .001), while the effect of Mindfulness was reduced to non-significance (from β = 0.169, p < .001 to β = 0.041, p = .079). A subsequent bootstrapped mediation analysis (5,000 resamples) was added to the manuscript to probe these relationships. It confirmed that the association with Mindfulness is fully explained through its relationship with Technophilia (i.e., full mediation). In contrast, Digital Competence maintains a significant direct effect on the Digital Centaur Scale, in addition to its indirect link. ]
Comments 5: [Sole reliance on self-reported DQ Index (1–10 scale) risks illusory superiority bias, particularly given: the minimal correlation between DQ scores and actual AI usage frequency (r=.08–.11).]
Response 5: [The low-to-moderate correlations between the Digital Centaur Scale and the use of AI for specific tasks (r = .09 to .20, p < .001) are likely indicative of what the scale is designed to measure. The Digital Centaur Scale, which demonstrated good reliability (Cronbach’s α = .80, M = 3.30, SD = 0.96), assesses a general affinity towards a "digital centaur" lifestyle through three attitudinal indicators: “How much do you like this person?”, “How similar are you to this person?”, and “How willing would you be to regularly communicate with someone who lives this kind of lifestyle?”. Respondents rated these items on a 5-point Likert scale from 1 (strongly disagree) to 5 (strongly agree). This conceptual focus on general affinity and identity, rather than specific behaviors, explains its modest correlation with concrete reports of AI usage frequency.
Nevertheless, the Digital Centaur Scale demonstrates a moderate correlation with Digital Competence (r = .30, p < .001), supporting its validity for use in this study. The scale, as operationalized by these indicators, primarily measures an affinity for or a preference for the Digital Centaur strategy. We have acknowledged this specific focus as a limitation in the manuscript. To address this in future research, we plan to develop additional methodological tools designed to assess concrete behavioral markers of being a Digital Centaur. Furthermore, refining the conceptualization of the construct across other psychological dimensions will help mitigate the potential influence of illusory superiority bias.]
Round 2
Reviewer 3 Report
Comments and Suggestions for Authors
THe authors have addressed all my concerns.